🔓 | **Open Peer Review** | Genomics and Proteomics | Research Article

# Is ICE hot? A genomic comparative study reveals integrative and conjugative elements as "hot" vectors for the dissemination of antibiotic resistance genes

Qi Zheng,[1] Liguan Li,[1] Xiaole Yin,[1] You Che,[1] Tong Zhang[1]

**ABSTRACT** The dissemination of antibiotic resistance genes (ARGs) driven by mobile genetic elements (MGEs), especially among pathogenic bacteria, is of increasing global concern. Different from other well-characterized MGEs, integrative and conjugative elements (ICEs) have been lacking a comprehensive understanding of their roles in ARG propagation across bacterial phylogenies. Through genomic study based on a large collection of bacterial complete genomes and further comparative analysis with two prominent MGEs to spread ARGs—conjugative plasmids and class 1 integrons, we, for the first time, demonstrated that ICEs are indeed overlooked "hot" vectors from the aspects of mobility and pathogenicity: (i) ICEs exhibited broader phylogenetic distribution among two dominant phyla with high ARG diversity and (ii) ARG-carrying ICEs were significantly enriched in potential human pathogens covering all the six "ESKAPE" species, of which some displayed typical co-occurrence patterns with ARGs and virulence factors. Moreover, this first genomic comparative study also deciphered the distinct ARG profiles harbored by these three essential MGE groups in terms of diversity and prevalence, with characteristic ARG preference to each MGE group. Overall, our findings concerning the MGE-specific performance for ARG transmission, in particular, the historically understudied ICEs, could shed light on control strategy optimization to antibiotic resistance crises.

**IMPORTANCE** Different from other extensively studied mobile genetic elements (MGEs) whose discoveries were initiated decades ago (1950s–1980s), integrative and conjugative elements (ICEs), a diverse array of more recently identified elements that were formally termed in 2002, have aroused increasing concern for their crucial contribution to the dissemination of antibiotic resistance genes (ARGs). However, the comprehensive understanding on ICEs' ARG profile across the bacterial tree of life is still blurred. Through a genomic study by comparison with two key MGEs, we, for the first time, systematically investigated the ARG profile as well as the host range of ICEs and also explored the MGE-specific potential to facilitate ARG propagation across phylogenetic barriers. These findings could serve as a theoretical foundation for risk assessment of ARGs mediated by distinct MGEs and further to optimize therapeutic strategies aimed at restraining antibiotic resistance crises.

**KEYWORDS** integrative and conjugative elements, conjugative plasmids, class 1 integrons, antibiotic resistance genes, pathogenic bacteria, comparative genomics

Induced by the selective pressure of inappropriately used antibiotics, antibiotic resistance genes (ARGs) have become a significant global concern since they may render therapeutic failure and further threaten human health and biosecurity (1, 2). It is evident that efficient ARG acquisition by pathogenic bacteria through horizontal gene

Address correspondence to Tong Zhang, zhangt@hku.hk.

The authors declare no conflict of interest.

See the funding table on p. 12.

transfer (HGT) acts as the essential driving force for the progressively severe threats of antibiotic resistance posed to public health (3, 4). HGT relies on the concerted activities of mobile genetic elements (MGEs) to facilitate the dissemination of ARGs and consequently confer the host bacteria with selective advantages under antibiotic pressure (5, 6).

Following the concept of HGT first proposed in 1947 (7), a wide variety of MGEs have gradually been discovered and recognized as genetic vectors to transfer ARGs (8). In general, MGEs promote DNA mobility either intracellularly including insertion sequences (IS) (9) and transposons (10) or intercellularly by mechanisms of transformation, transduction, and conjugation. As for the conjugation mechanism, conjugative plasmids play a pivotal role in the development of bacterial antibiotic resistance, particularly propagating ARGs among nosocomial pathogens (11). Apart from the above mobility mechanisms, integrons as non-autonomous elements relying on other MGEs to conduct intracellular or intercellular transmission are essentially versatile repositories to accumulate exogenous genes such as ARGs (12). Indeed, ARG-encoding class 1 integrons—the central components of integrons, have successfully invaded diverse pathogenic taxa of clinical importance and further facilitate antibiotic resistance spread (13).

Different from the aforementioned MGEs whose discoveries were initiated decades ago (1950s–1980s), integrative and conjugative elements (ICEs), a diverse array of more recently identified elements that were formally termed in 2002 (14), have aroused increasing concern for their crucial contribution to the dissemination of ARGs (15). ICEs share the same conjugation machinery as conjugative plasmids to conduct intercellular transmission; however, unlike plasmids that are extrachromosomal, ICEs can be integrated into the bacterial chromosome by reversible site-specific recombination (16). Moreover, the structure of ICEs is typically modular, that is, genes of similar functions are clustered together. They comprise three core modules: recombination, conjugation, and regulation (17). According to the conjugation module, ICEs are classified into two categories—T4SS-type ICEs transferring as single-stranded DNA among the majority of bacteria and AICEs transferring as double-stranded DNA only within the phylum *Actinobacteria* (18). Except for the three backbone-like core modules, ICEs also encompass various accessory genes including ARGs and other functional genes like virulence factors (VFs), thus making ICEs vital drivers for bacterial adaptation and evolution. In fact, the discovery of ICEs could be traced back to studies on these accessory genes' conjugative transposition between cells in the absence of plasmids (16).

Previous research mainly concentrated on well-characterized model ICEs to explore their association with ARGs from different aspects. A major focus was to interpret the connection between infectious disease emergence and certain ARG-carrying ICEs such as scarlet fever outbreaks caused by ICE-*emm*12 encoding genes resistant to tetracycline and macrolide located on *Streptococcus pyogenes* isolates (19) or neonatal infection rise attributed to *tetM*-harboring ICEs of Tn*916* family hosted by *Streptococcus agalactiae* clones (20). Moreover, fundamental light was shed on the underlying mechanisms of ARG transmission by ICEs (21, 22) and the discovery of novel ICEs like ICE*Apl*Chn1 under SXT/R391 family identified from multidrug-resistant pathogen *Actinobacillus pleuropneumoniae* (23), as well as novel ARGs like *tetX6* embedded in another SXT/R391 member, ICE*Pgs6*Chn1, from tetracycline-resistant pathogen *Proteus* sp. (24). Additionally, previous efforts by scientists have also provided further insight into the potential of ICEs for ARG propagation among microbial communities from a wide range of environments, such as human and animal tissues (25–27), livestock farms (28), aquatic ecosystems (29, 30), as well as lab-scale bioreactors (31). Yet, as these studies were limited to specific ICE families/host bacteria/environmental habitats, we are still lacking a comprehensive understanding of ICEs with respect to their roles in acquiring and disseminating ARGs across the bacterial tree of life, especially those pathogenic bacteria of clinical concern.

Indeed, it has been argued that ICEs are largely overlooked as significant vectors of ARGs (15, 32).

Given the current knowledge on ICEs, we pose two questions: (i) are ICEs "hot" MGEs in terms of ARG dissemination among bacteria? and (ii) are ICEs distinct contributors to ARG spread from other MGEs? To address these questions, we exploited the exponentially expanding database of bacterial complete genomes (33) to decipher the roles of ICEs in ARG propagation across bacterial phylogenies through comparative analysis with two representative MGEs—conjugative plasmids and class 1 integrons as prominent elements for ARG transmission, especially in nosocomial settings. We, for the first time, demonstrated that ICEs are "hot" vectors in the acquisition and dissemination of ARGs with disparate performance from plasmids and integrons. The distinct ARG profiles harbored by these three essential MGEs are of great significance to illuminate MGE-specific potential to facilitate ARG propagation across phylogenetic barriers, which also provides a scientific basis for curbing ARG spread mediated by different MGEs.

## RESULTS

### Distinct ARG profiles across the three MGE groups

Totally, 3,761 T4SS-type ICEs, 403 AICEs, 3,180 conjugative plasmids, and 1,008 class 1 integrons were extracted from the complete genome database, residing on 17%, 1%, 13%, and 5% of these bacterial genomes, respectively. Among them, 560 (15%) T4SS-type ICEs, 1,352 (43%) conjugative plasmids, and 868 (86%) class 1 integrons carried ARGs (Table S2). Since the retrieved AICEs did not harbor any ARGs, they were excluded from the downstream analysis. Besides, the terminology is henceforth simplified as ICEs for T4SS-type ICEs, plasmids for conjugative plasmids, and integrons for class 1 integrons. ICEs (108.8 ± 75.3 kb) and plasmids (130.0 ± 110.0 kb) were evidently longer than integrons (4.5 ± 1.6 kb). In addition, the ARG-bearing MGEs also exhibited a broad spectrum of size distribution (Fig. S1A). Moreover, the ARG number per plasmid (average value 4.4) was significantly larger than that of ICEs (2.2) as well as integrons (1.8) ($P$-value < 0.01, Benjamini-Hochberg [B-H] corrected Mann-Whitney test). To be more specific, the majority of ICEs (88%) and integrons (98%) encoded 1–4 ARGs, and a few of the ICEs carried more than five ARGs, which was, however, rarely observed in integrons. Compared with ICEs and integrons, plasmids were inclined to harbor more ARGs, especially some plasmids even encoded over 10 ARGs (Fig. S2).

ICEs and plasmids possessed more diverse ARGs including 16 and 13 ARG types composed of 139 and 147 subtypes, respectively, compared to 8 ARG types of 76 subtypes harbored by integrons (Table S3). Additionally, the three MGEs demonstrated distinct ARG profiles and only 34 ARG subtypes (out of a total of 247 subtypes) were shared among them (pairwise Jaccard indices ≤ 0.30; Fig. 1B). Further exploration of the association between ARGs and MGEs revealed that certain ARG types exhibited characteristic preference to specific MGE groups (Fig. 1A). For instance, aminoglycoside-resistant ARGs were prevalent among all the three MGEs. On the contrary, ARGs resistant to tetracycline, beta-lactam, and trimethoprim were dominant on one of the three MGEs. In particular, tetracycline-resistant ARGs were detected from more than half (57%) of the ARG-carrying ICEs, accounting for roughly 28% of the total ARGs harbored by them. Furthermore, tetracycline-resistant ARGs on ICEs displayed rich diversity since they covered nearly half of their relevant subtypes in the SARG database (19 out of 43). Different from ICEs, the most abundant ARG types possessed by plasmids and integrons were those against beta-lactam (25% within plasmids' ARG reservoir) and trimethoprim (27% within integrons'), with wide spread of 74% and 48% among these two ARG-encoding MGE groups, respectively. Similar to ARG types, several ARG subtypes also presented a typical tendency toward ICEs and integrons (Fig. 1C). For example, *tetM* resistant to tetracycline exploiting antibiotic target protection mechanism (34) was enriched in ICEs as this gene was prevalent among 35% of ARG-bearing ICEs, dominating the largest proportion (15%) in ICEs' ARG reservoir. By contrast, *tetM* was only detected on 2% of ARG-carrying plasmids and was even absent

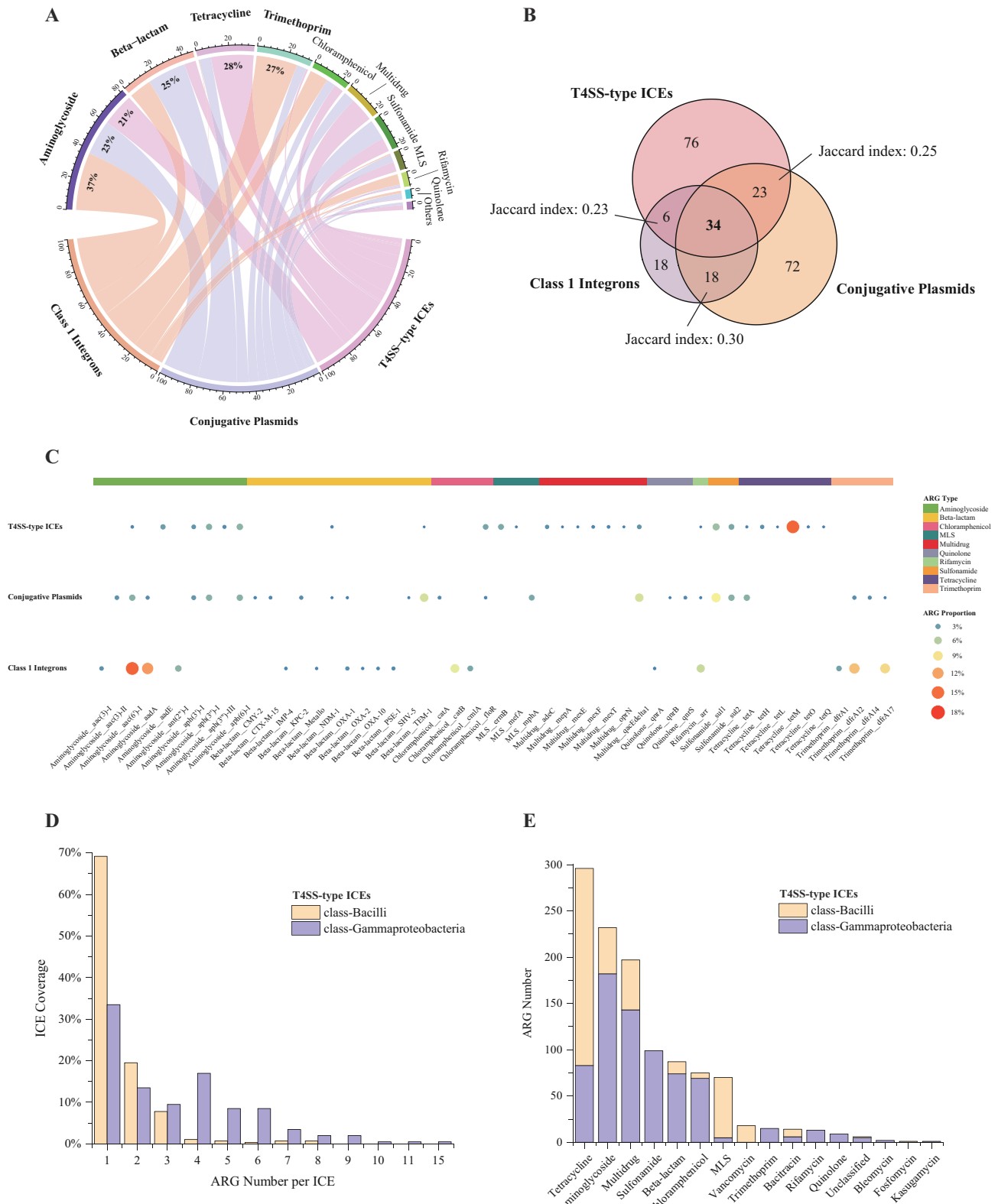

**FIG 1** ARG profiles of the three MGEs—T4SS-type ICEs, conjugative plasmids, and class 1 integrons (simplified as ICEs, plasmids, and integrons). (A) Chord diagram presents the proportion of different ARG types harbored by the three MGEs. ARG types of minor proportion among all the three MGEs with summation < 1.5% were merged as "Others". MLS, macrolide-lincosamide-streptogramin. (B) Venn diagram demonstrates the number of shared and unique ARG subtypes across the three MGEs. Jaccard index was calculated as the ratio of intersection over union between two MGEs pairwise. (C) Heat map displays ARG composition (Continued on next page)

**FIG 1** (Continued)

at the subtype level of the three MGEs. For each MGE, only ARG subtypes of abundance over 1% are displayed, with aggregated abundance higher than 70%. The ARG subtype proportion is indicated by circle size and color, and their categorized ARG types are annotated by the top color strip. (D) Bar chart shows the coverage distribution (divided by the ARG-carrying ICEs belonging to each class separately) of ICEs encoding different ARG numbers in the two major classes—*Bacilli* and *Gammaproteobacteria*. (E) Bar chart exhibits the ARG number of different types possessed by ICEs in the two major classes.

on integrons. Other examples were *aac*(6')-*I* and *aadA* with aminoglycoside resistance as well as *dfrA* with trimethoprim resistance, both dominant in the MGE group of integrons. Instead, there was no obvious enrichment of specific ARG subtypes on plasmids.

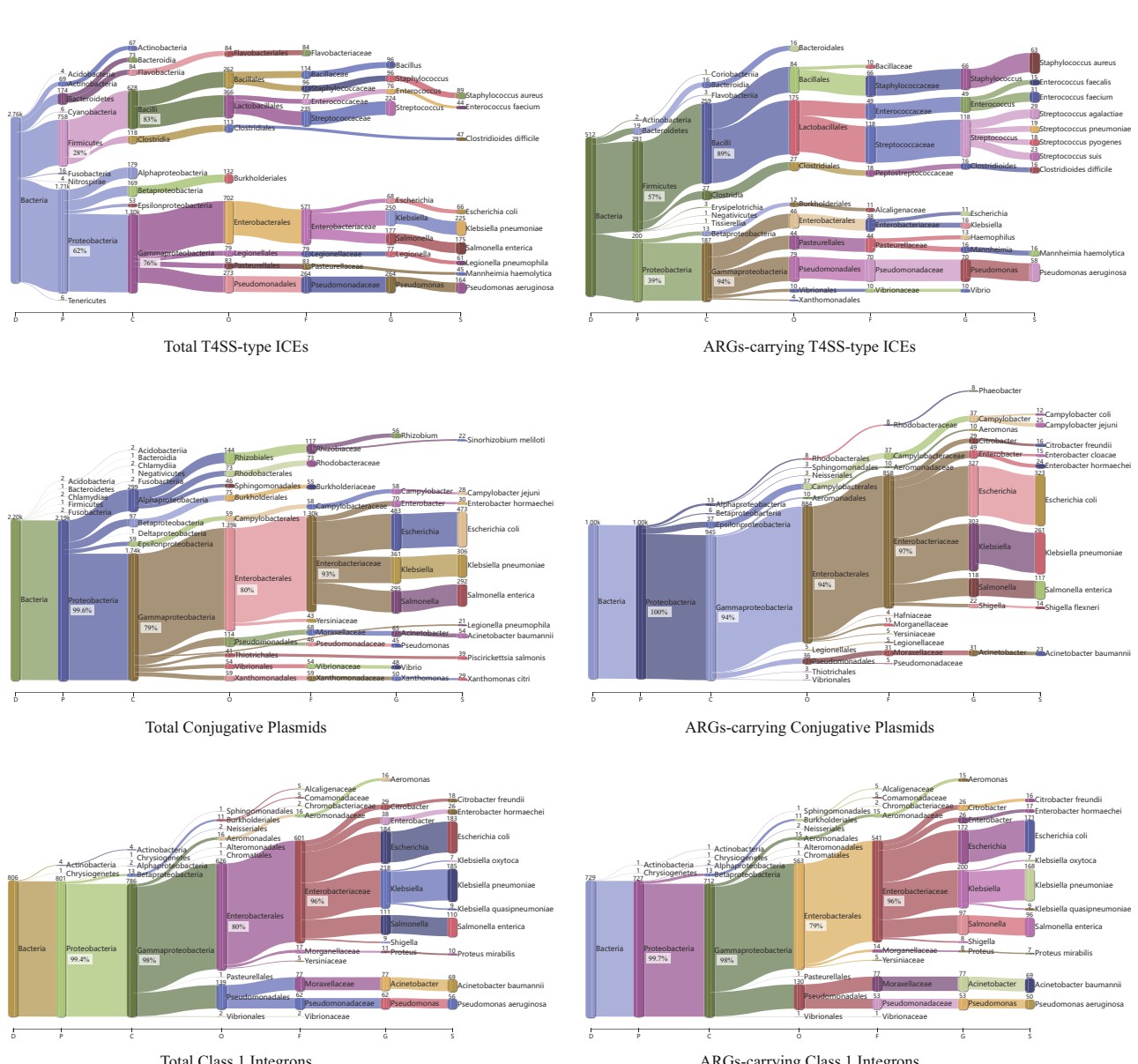

**FIG 2** Phylogenetic distribution of bacteria hosting the ICEs, plasmids, and integrons, as well as their ARG-carrying ones. Marked integer above the node denotes the genome number subject to a certain taxonomic level, and marked percentage refers to the proportion occupying the total genomes at its upper level. Taxonomic levels are labeled on the bottom of each panel: D, domain; P, phylum; C, class; O, order; F, family; G, genus; and S, species. Only top 10 clades are presented under each taxonomic level.

## Phylogenetic conservation of MGEs' host bacteria

ICEs exhibited broader phylogenetic distribution compared with plasmids and integrons (Fig. 2). Specifically, both plasmids and integrons as well as their ARG-carrying ones were dominant in *Proteobacteria* (> 99%). Unlike this highly phylogenetic conservation, even though ICEs were also mainly located in *Proteobacteria* (62%), another phylum, *Firmicutes*, existed to host a considerable amount of ICEs (28%). Remarkably, the opposite was observed for ARG-bearing ICEs where the primary host phylum converted to *Firmicutes* (57%), followed by *Proteobacteria* (39%). Moreover, at a lower taxonomic level under *Proteobacteria*, all the three MGEs were distributed unevenly across classes, i.e., they were predominantly embedded in *Gammaproteobacteria* rather than other classes. Likewise, under another dominant phylum harboring ICEs—*Firmicutes*, *Bacilli* acted as the major class. As for plasmids and integrons, the phylogenetic conservation was further extended to the family level under the class *Gammaproteobacteria*, that is, both of them were restricted to the family *Enterobacteriaceae*. It is notable that similar phenomena regarding heterogeneous phylogenetic distribution were also observed from the ARG-encoding ones of the three MGE groups.

The two major classes hosting ARG-carrying ICEs—*Bacilli* and *Gammaproteobacteria* displayed distinct distribution profiles in terms of ARG abundance and diversity (Table S4). On the one hand, although *Gammaproteobacteria* was the second dominant host, it harbored more abundant ARGs (54% of the total ARGs encoded by ICEs) than the most dominant host, *Bacilli* (33%). Besides, ICEs in *Gammaproteobacteria* also tended to carry multiple ARGs, whereas the majority (69%) of ICEs in *Bacilli* were one-ARG carriers (Fig. 1D). Meanwhile, the ARG-bearing ICEs harbored by *Gammaproteobacteria* (183.8 ± 132.4 kb) were apparently larger than the *Bacilli*-harbored ones (85.9 ± 44.7 kb) (Fig. S1B). On the other hand, *Gammaproteobacteria*-hosted ICEs possessed more diverse ARGs (93 subtypes) over two times compared to those located in *Bacilli* (43 subtypes). Notably, pretty limited ARG subtypes were shared between the two classes (Jaccard index 0.05). In fact, specific ARGs encoded by ICEs were distributed heterogeneously among these two classes. For instance, tetracycline-resistant ARGs were primarily carried by ICEs in *Bacilli*, especially their core member *tetM* as 88% of *tetM*-encoding ICEs were detected in this phylogeny (Fig. S5). By contrast, ICEs associated with ARGs resistant to aminoglycoside, multidrug, sulfonamide, beta-lactam, and chloramphenicol were more likely to be hosted by *Gammaproteobacteria* (Fig. 1E).

## Enrichment of ARG-carrying MGEs in pathogens

Consistently observed across the three MGEs encoding ARGs, all of them were significantly enriched in potential human pathogens (*P*-value < 0.01, B-H corrected Fisher's exact test). In detail, 41% of the bacterial species hosting ARG-carrying ICEs, as well as 35% of those harboring ARG-bearing plasmids or integrons, were pathogenic species, nearly six times compared to the proportion of pathogenic species within the complete genome database (7%) (Fig. 3A). Moreover, these ARG-encoding MGEs located on bacterial pathogens possessed the majority (approximately 80%) of their ARG reservoirs. It is noteworthy that although the three MGE groups hosted by non-pathogens encoded much less abundant and diverse ARGs than their pathogen-hosted ones, the distribution profiles with respect to major ARG types and subtypes across MGEs on pathogens and non-pathogens all exhibited significant similarity (*P*-value > 0.1, paired *t*-test; Fig. 3B; Table S5).

In general, the total 48 pathogenic species hosting ARG-carrying ICEs were distributed broadly among two phyla—*Firmicutes* (mainly within the class *Bacilli*) and *Proteobacteria* (mainly within the class *Gammaproteobacteria*) (Fig. S6). It is of particular concern that these bacterial pathogens covered all the six "ESKAPE" species as ubiquitous pathogens causing life-threatening infections in healthcare settings (35). Especially, two "ESKAPE" members *Pseudomonas aeruginosa* (36) and *Staphylococcus aureus* (37) both harbored 15% of ARG-bearing ICEs located on pathogens, and their proportions were much higher than other species (Fig. 3C). Furthermore, *P. aeruginosa* possessed the

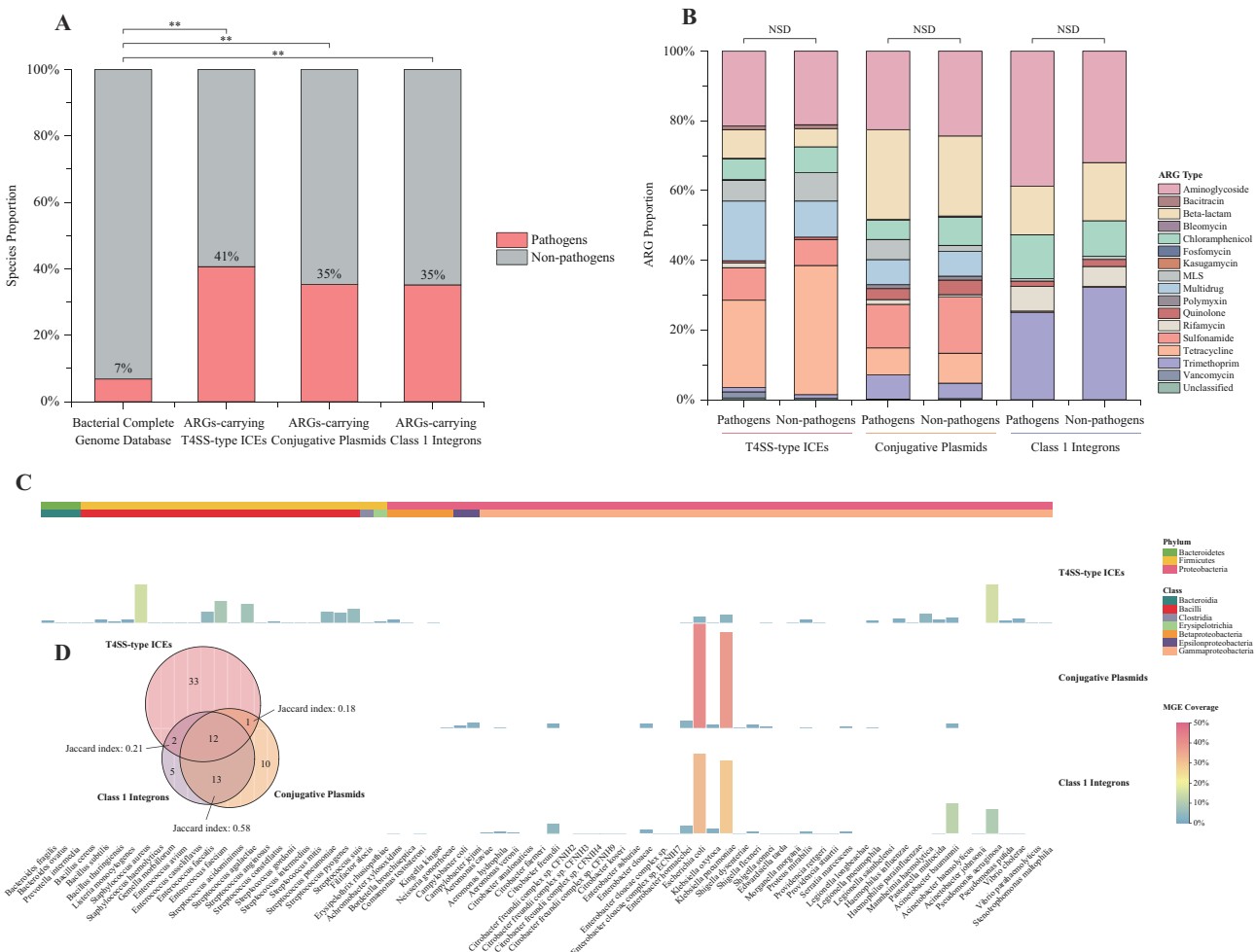

**FIG 3** Enrichment of ARG-carrying MGEs in pathogens. (A) Bar chart presents the pathogenic proportion normalized to bacterial species level of the ARG-carrying ICEs, plasmids, and integrons, as well as the complete genome database. *P*-value was evaluated via B-H corrected Fisher's exact test, and **$P$-value < 0.01. (B) Bar chart exhibits ARG composition at the type level of the three MGEs hosted by pathogens and non-pathogens. *P*-value was evaluated via paired *t*-test; NSD, no significant difference (*P*-value > 0.1). (C) Heat map displays coverage distribution of the three ARG-encoding MGEs harbored by various pathogenic species. The MGE coverage (divided by ARG-encoding MGEs located on bacterial pathogens) is indicated by bar height and color, and their host bacteria are annotated to phylum and class levels by the top color strips. (D) Venn diagram demonstrates the number of shared and unique pathogenic species hosting the three ARG-bearing MGEs.

most abundant ARGs (20%) with the richest diversity (36 ARG subtypes) from ICEs' reservoir (Table S6). The typical multidrug-resistant mutant of *mexEF-oprN* and *mexT* was prevailing among one-fifth of ARG-encoding ICEs embedded in *P. aeruginosa*. Indeed, this multidrug-resistant combination is frequently detected from *P. aeruginosa* to overproduce the active efflux system, i.e., the *mexEF-oprN* efflux pump is overexpressed by its activator *mexT* mutation to confer multidrug resistance (38). Besides, *tetM* appeared predominant on ICEs that were hosted by another major ICE-carrying species *S. aureus* (39) (Table S7). Contrary to ICEs, the pathogenic species harboring ARG-bearing plasmids (36 species) and integrons (32 species) were both conserved in *Proteobacteria* (mainly within the family *Enterobacteriaceae*) (Fig. S6); meanwhile, most of these host species were shared between the two MGEs (Jaccard index 0.58; Fig. 3D). Remarkably, the majority of ARG-carrying plasmids (79%) and integrons (60%) were equally possessed by two pivotal enteric pathogens *Escherichia coli* and *Klebsiella pneumoniae* (40) (Fig. 3C). Moreover, these two MGE groups located on the two dominant species presented the most abundant and diverse ARG profiles in line with their overall pictures of ARG distribution respectively (*P*-value > 0.1, paired *t*-test; Tables S6 and S7).

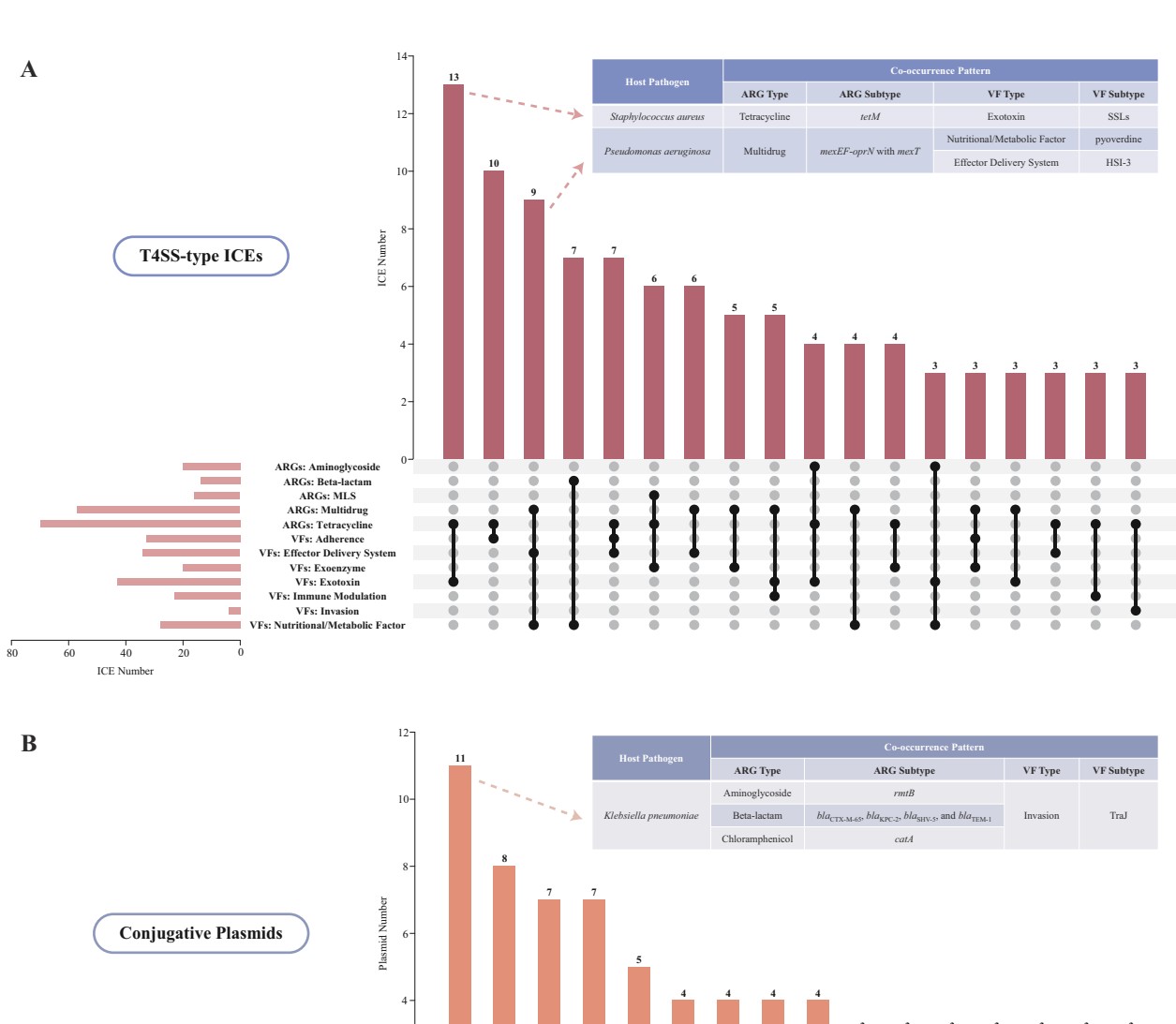

**FIG 4** Typical co-occurrence patterns of ARGs and VFs harbored by ICEs (A) and plasmids (B). UpSet plot exhibits the co-occurrence patterns of ARGs and VFs at the type level, and only patterns appearing on more than two MGE numbers are displayed. Combination of intersection points as black dots with connecting line depicts the type-level co-occurrence pattern, and vertical bar chart on the top demonstrates the MGE number harboring certain patterns. Meanwhile, horizontal bar chart on the left demonstrates the MGE number with co-occurrence patterns encoding certain ARG or VF types. The detailed genetic co-occurrence patterns at the subtype level with the highest abundance on these two MGEs hosted by their dominant pathogenic species are additionally plotted in corresponding tables.

## Co-occurrence of ARGs and VFs

Apart from the defense mechanism of ARGs, the three MGEs also act as potential vectors for attack system of VFs, thus equipping their host bacteria with strong

competitive edges under antibiotic treatment. The co-occurrence of ARGs and VFs on MGEs hosted by opportunistic pathogens is of definitely high risk to public health given their enhanced pathogenicity and antibiotic resistance with mobility (41). Therefore, VF profiles especially those co-existing with ARGs harbored by the three MGE groups were further explored.

VFs appeared more prevalent on ICEs in contrast with the other two MGEs, i.e., 20% of ICEs carried VFs versus 14% of plasmids and none of integrons. Additionally, VF subtypes encoded by ICEs (132 subtypes) were more than twice the diversity of those subtypes on plasmids (52 subtypes). In detail, ICEs and plasmids possessed distinct VF profiles (Jaccard index 0.12): ICEs tended to harbor yersiniabactin (classified as VF category of nutritional/metabolic factor) and Vi antigen (immune modulation), whereas plasmids were prone to VFs of aerobactin and salmochelin siderophore (nutritional/metabolic factor), as well as Pef (adherence) (Table S8). Interestingly, contrary to the phylogenetic distribution of ARG-carrying ICEs (refer to the section "Phylogenetic conservation of MGEs' host bacteria"), ICEs encoding VFs were predominantly hosted by the class *Gammaproteobacteria* rather than *Bacilli* (Fig. S7A).

Through coupling the distribution profiles of ARGs and VFs, it is remarkable that the majority of ICEs (92%) and plasmids (82%) with co-occurrence of these two functional genes were located on potential human pathogens. Furthermore, certain co-occurrence patterns were detected from the two MGE groups, in particular, for those hosted by their dominant pathogenic species. Specifically, the primary ARG *tetM* embedded on *S. aureus*-harbored ICEs preferred to co-exist with their most abundant VF SSLs (under the VF category of exotoxin) (42). Regarding ICEs harbored by *P. aeruginosa*, the multidrug-resistant combination of *mexEF-oprN* efflux pump regulated via *mexT* activator was more likely to co-occur with two major VFs of pyoverdine (nutritional/metabolic factor) (43) and HSI-3 (effector delivery system) (44) (Fig. 4A; Table S9). As for plasmids located on *K. pneumoniae*, the most widespread VF TraJ (invasion) was inclined to co-exist with a variety of ARG types—aminoglycoside (*rmtB*), beta-lactam ($bla_{CTX-M-65}$, $bla_{KPC-2}$, $bla_{SHV-5}$, and $bla_{TEM-1}$), as well as chloramphenicol (*catA*) (Fig. 4B; Table S9) (45, 46).

## DISCUSSION

Different from extensively characterized plasmids and integrons, the comprehensive profile of ICEs for promoting ARG dissemination across phylogenetic tree is still blurred. Through genomic study based on a large collection of bacterial complete genomes, we, for the first time, systematically explored the ARG profile as well as the host range of ICEs by comparison with the two key MGEs to spread ARGs—plasmids and integrons (Table 1). We found that the three MGE groups possessed distinct ARG profiles in that certain ARGs presented a typical preference for specific groups. Meanwhile, both ICEs and plasmids harbored richer ARG diversity, whereas integrons exhibited higher ARG prevalence since the majority of them carried ARGs. It is noteworthy that all the three ARG-encoding MGEs were significantly enriched in potential human pathogens, which pose severe threats to public health. In addition, this first genomic comparative study also revealed that ICEs are indeed overlooked "hot" vectors to facilitate ARG propagation from aspects of mobility and pathogenicity: (i) ICEs demonstrated broader phylogenetic distribution among two major phyla with high ARG diversity and (ii) pathogenic bacteria hosting ARG-carrying ICEs covered all the six "ESKAPE" species, of which some displayed typical co-occurrence patterns with ARGs and VFs.

The integrons identified in our study contain typical structures of clinical form, that is, they essentially exist in nosocomial contexts under intensive antibiotic selective pressure (47), which might explain why ARGs were prevalent among this MGE group. By contrast, the reasons for high ARG diversity appearing on ICEs and plasmids probably lie in their large sizes (18), as well as versatile interactions with other MGEs like IS, transposons, or integrons (8, 16). Except for the common feature of high ARG diversity, these two MGEs also possessed distinct performance: ICEs exhibited broader phylogenetic distribution, while plasmids were more likely to harbor multiple ARGs. This difference

**TABLE 1** Comparative summary for distinct profiles of the three MGEs concerning ARG dissemination[a]

|  | T4SS-type ICEs | Conjugative plasmids | Class 1 integrons |
|---|---|---|---|
| Size (kb) | Long (108.8 ± 75.3) | Long (130.0 ± 110.0) | Short (4.5 ± 1.6) |
| ARG diversity | High (16 types, 139 subtypes) | High (13 types, 147 subtypes) | Medium (8 types, 76 subtypes) |
| ARG prevalence (%) | Low (15) | Medium (43) | High (86) |
| Multiple ARGs per MGE | Medium | High | Low |
| Shared ARG type | Aminoglycoside | Aminoglycoside | Aminoglycoside |
| ARG inclination (type level) | Tetracycline | Beta-lactam | Trimethoprim |
| ARG inclination (subtype level) | *tetM* (tetracycline) | ND | *aac*(6′)-*I* (aminoglycoside) |
|  |  |  | *aadA* (aminoglycoside) |
|  |  |  | *dfrA* (trimethoprim) |
| Phylogenetic distribution | Two phyla | One phylum | One phylum |
| Phylogenetic conservation | Yes | Yes | Yes |
| Enrichment in pathogens | Yes | Yes | Yes |
| Dominant pathogenic species | *Pseudomonas aeruginosa* | *Escherichia coli* | *Escherichia coli* |
|  | *Staphylococcus aureus* | *Klebsiella pneumoniae* | *Klebsiella pneumoniae* |
| Co-occurrence pattern of ARGs and VFs | Yes | Yes | ND |

[a]ND, not detected.

seems to be in accordance with their disparate transmission dynamics (15). Generally, ICEs are more stable than plasmids as they hold an obvious dualistic mode of life termed "bistability". In addition to the horizontal transmission between cells, ICEs can also maintain themselves within cells by integrating into and replicating along with the host chromosome for vertical transmission, which is typically immune to segregational loss. Instead, most plasmids suffer from segregational loss during cell division. Besides, ICEs prefer to integrate site specifically and mostly at conserved chromosomal target sites; therefore, they could broaden the host range with stable maintenance (48). Our results conformed to this fundamental principle that ICEs spread widely among two dominant phyla, whereas plasmids were restricted to a single phylum. Furthermore, the integrated state of ICEs indicates their quiescent occurrence with constitutive repression for the conjugation machinery (49). In fact, there existed phylogenetic conservation at a lower taxonomic level under ICEs' two phyla; meanwhile, these two major classes hosting ARG-carrying ICEs presented distinct distribution profiles, which implies the biological and ecological limitations of ICEs in acquiring and disseminating ARGs (50). On the other hand, plasmids are of increased HGT capacity compared to ICEs, with higher transfer rates as well as gene exchange frequencies (51). This may interpret the featured performance of plasmids to harbor massively multiple ARGs. Moreover, the active exchange network of ARGs mediated by plasmids was detected across genera under the phylum *Proteobacteria* in a previous study (52).

Remarkably, *tetM* was a particular highlight on ICEs, given not only the expansive spread among this MGE group but also the typical co-occurrence with VFs to strengthen selective advantages for host bacteria under antibiotic treatment. This ARG is recognized to reside on the "super-mobile" ICEs such as Tn*916*-like elements (53, 54). Tn*916*-like elements exhibit low integration specificity and insert preferentially into AT-rich sites for their efficient propagation across a wide variety of hosts (55, 56), which could be employed to elucidate the prominent existence of *tetM* on ICEs.

Overall, exploration of the MGE-specific performance in ARG acquisition and dissemination, especially shedding light on the historically understudied ICEs, could serve as a theoretical foundation not only for risk assessment of ARGs mediated by these distinct MGEs but also to optimize therapeutic strategies aimed at restraining antibiotic-resistance crises (57, 58). However, we have to admit that the conclusions drawn from this study might be biased to some extent, i.e., the analyzed bacterial complete genome database possesses inherent bias toward certain taxa, such as common species under the phyla *Proteobacteria* and *Firmicutes*, which are relatively easier to be cultured and then investigated, causing the exclusion of diverse not-yet-cultured species in nature

(Fig. S3). Besides, future work is needed to experimentally evaluate and compare the HGT potential of the three MGEs, in particular, to illuminate the underlying molecular mechanisms behind the characteristic ARG preference of each MGE group.

## MATERIALS AND METHODS

### Bacterial complete genome collection

A total collection of 16,364 bacterial complete genomes covering 48 phyla were downloaded from the NCBI genome database (59) in GenBank format on 7 January 2020. The taxonomic lineages of these downloaded genomes were retrieved from the NCBI taxonomy database (60) via TaxonKit (61) according to the specific TaxID associated with each genome assembly accession number. The detailed genomic metadata used in this study are summarized in Table S1.

### MGE extraction

The bacterial genome sequences were first divided into chromosomes and plasmids by keyword retrieval against their annotation information in GenBank files. ICEs were detected from the chromosome sequences and further classified as T4SS-type ICEs or AICEs using ICEfinder (17), which identifies the signature backbone modules of these two ICE categories, i.e., elements carrying integrase, *oriT*, relaxase, T4CP, and T4SS are T4SS-type ICEs, while those encoding integrase, replication initiator, and translocation proteins are AICEs. Flanking direct repeats were also detected as the defined boundaries of ICEs. These detected ICEs were double confirmed by a systematic genomic island classification tool AtollGen-CLI (62) according to the combination of their mobility signature proteins. In addition, ICEs extracted from the phylum *Firmicutes* were verified through ICEscreen (63) with reference to the composite structures of signature proteins dedicated to ICEs located in *Firmicutes*. The identified plasmid sequences were subsequently categorized via Plascad (52) based on their genetic machinery for DNA transfer into conjugative plasmids carrying relaxase, T4CP, and T4SS, mobilizable plasmids encoding only relaxase, as well as non-mobilizable plasmids missing all these functional genes. Among the aggregated 14,813 plasmid sequences, roughly half (51%) were classified as non-mobilizable plasmids, followed by mobilizable plasmids (28%) and conjugative plasmids (21%). The phylogenetic distribution of their host bacteria is demonstrated in Fig. S4. Besides, according to integrons' evolutionary history, the class 1 integrons recovered from clinical contexts have evolved into a typical structure with *intI1* as 5′-conserved segment (5′-CS), *attC* related to gene cassettes, and fused genes of *qacEΔ1*/*sul1* as 3′-conserved segment (3′-CS) (64). The three structural features were utilized by I-VIP (65) to extract these clinical class 1 integrons from both bacterial chromosomes and plasmids.

### ARG and VF retrieval

Protein-coding regions on the three MGEs were further subject to BLASTP search (66) against the structured ARG database (SARG v2.2) (67) and virulence factor database (VFDB core data set) (68) to explore the functional genes of ARGs and VFs harbored by them. In detail, SARG v2.2 incorporates 1,244 ARG subtypes of genetic names attributed to 24 ARG types indicating antibiotic classes to which these genes confer resistance. Similarly, VFDB in the core data set version encompasses representative VFs of experimental verification, covering 536 VF subtypes with genetic names under 14 VF types referring to their pathogenic mechanism categories. The criteria of BLASTP search were *e*-value ≤ 1e−5, hit similarity ≥ 90%, and alignment length coverage ≥ 80% for ARG retrieval, whereas *e*-value ≤ 1e−5, hit similarity ≥ 80%, and alignment length coverage ≥ 90% for VF retrieval (69). Since ARGs of *qacEΔ1* and *sul1* act as signature structures for clinical class 1 integrons, these two ARG subtypes were excluded from the downstream analysis of this MGE group. Additionally, we detected a few ARG-encoding class 1

integrons embedded in T4SS-type ICEs and conjugative plasmids; nevertheless, these integron-borne ARGs only accounted for 3% and 10% within ICE- and plasmid-borne ARGs, respectively. Besides, the ARG profiles of integrons located on these two MGEs both exhibited significant similarity to the overall picture of ARG-carrying integrons (*P*-value > 0.1, paired *t*-test; Table S10), namely, there existed no specific enrichment of integron-borne ARGs among these two MGE groups. Given the minor effect, the ARG reservoirs of ICEs and plasmids investigated in this study included these integron-borne ARGs.

## Pathogenicity analysis

The potential pathogenicity of MGEs' host bacteria was identified by matching their taxonomic annotation with a well-curated database containing 538 recognized bacterial species of human pathogens (70). The enrichment of ARG-bearing MGEs in potential human pathogens was statistically analyzed via Fisher's exact test compared to the pathogenic proportion of the complete genome database, with *P*-value further adjusted using B-H correction for false discovery rate control in multiple comparisons. Notably, in order to avoid the database bias caused by redundant sequencing of genomes especially for bacterial pathogens, the pathogenic proportion was normalized to species level, that is, every species was equally counted as one regardless of the genome number within this species.

## ACKNOWLEDGMENTS

This work was supported by the Theme-based Research Scheme of Hong Kong (T21-705/20-N) from the Research Grants Council (RGC).

We express sincere gratitude to The University of Hong Kong (HKU) for postgraduate scholarship and postdoctoral fellowship. We also greatly appreciate the technical assistance of computational resources from Dr. Ruibang Luo.

Q.Z. and T.Z. conceived and designed this research study. Q.Z. analyzed the data and wrote the manuscript. L.L. provided valuable advice on data analysis and manuscript writing. T.Z. guided and supervised the study. All authors contributed to revising the manuscript and approved the submitted version.

## AUTHOR AFFILIATION

[1]Department of Civil Engineering, Environmental Microbiome Engineering and Biotechnology Laboratory, Center for Environmental Engineering Research,The University of Hong Kong, Hong Kong, China

## AUTHOR ORCIDs

Qi Zheng  http://orcid.org/0000-0001-7155-1203

## FUNDING

| Funder | Grant(s) | Author(s) |
| --- | --- | --- |
| Research Grants Council, University Grants Committee (研究資助局) | T21-705/20-N | Qi Zheng |
| Research Grants Council, University Grants Committee (研究資助局) | T21-705/20-N | Liguan Li |
| Research Grants Council, University Grants Committee (研究資助局) | T21-705/20-N | Xiaole Yin |
| Research Grants Council, University Grants Committee (研究資助局) | T21-705/20-N | You Che |
| Research Grants Council, University Grants Committee (研究資助局) | T21-705/20-N | Tong Zhang |

## ADDITIONAL FILES

The following material is available online.

### Supplemental Material

**Figure S1 (mSystems00178-23-s0001.pdf).** Size distribution of the three MGEs and their ARG-carrying ones (A), as well as T4SS-type ICEs across the two major classes—*Bacilli* and *Gammaproteobacteria* (B).

**Figure S2 (mSystems00178-23-s0002.pdf).** Coverage distribution of the three MGEs encoding different ARG numbers among their total ARG-carrying ones correspondingly.

**Figure S3 (mSystems00178-23-s0003.pdf).** Phylogenetic distribution of the NCBI bacterial complete genome database (based on genome number).

**Figure S4 (mSystems00178-23-s0004.pdf).** Phylogenetic distribution of bacteria hosting the total 14,813 plasmids as well as their three categories—conjugative, mobilizable, and non-mobilizable plasmids (based on genome number).

**Figure S5 (mSystems00178-23-s0005.pdf).** Phylogenetic distribution of bacteria hosting the T4SS-type ICEs encoding *tetM* resistant to tetracycline (based on genome number).

**Figure S6 (mSystems00178-23-s0006.pdf).** Phylogenetic distribution of pathogenic species hosting the three ARG-carrying MGEs (based on species number).

**Figure S7 (mSystems00178-23-s0007.pdf).** Phylogenetic distribution of bacteria hosting the two MGEs that carry ARGs and VFs—T4SS-type ICEs (A) and conjugative plasmids (B) (based on genome number).

**Supplemental legends (mSystems00178-23-s0008.docx).** Legends for supplemental figures and tables.

**Supplemental tables (mSystems00178-23-s0009.xlsx).** Tables S1 to S10.

### Open Peer Review

**PEER REVIEW HISTORY (review-history.pdf).** An accounting of the reviewer comments and feedback.

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
