## [Reviewer comments · mSystems]

Is ICE Hot: A genomic comparative study reveals integrative and conjugative elements (ICEs) as “hot” vectors for the dissemination of antibiotic resistance genes

Qi Zheng, Liguan Li, Xiaole Yin, You Che, and Tong Zhang

Corresponding Author(s): Tong Zhang, The University of Hong Kong

Review Timeline:

Submission Date:	February 21, 2023
Editorial Decision:	April 27, 2023
Revision Received:	July 5, 2023
Editorial Decision:	August 12, 2023
Revision Received:	October 11, 2023
Accepted:	October 14, 2023

Editor: Li Cui

Reviewer(s): Disclosure of reviewer identity is with reference to reviewer comments included in decision letter(s). The following individuals involved in review of your submission have agreed to reveal their identity: Xinli An (Reviewer #3)

Transaction Report:

DOI: <https://doi.org/10.1128/msystems.00178-23>

April 27, 2023

Prof. Tong Zhang
The University of Hong Kong
Hong Kong
China

Re: mSystems00178-23 (Is ICE Hot: A genomic comparative study reveals integrative and conjugative elements (ICEs) as "hot" vectors for the dissemination of antibiotic resistance genes)

Dear Prof. Tong Zhang:

Thank you for submitting your manuscript to mSystems. We have completed our review and I inform you that, in principle, we expect to accept it for publication in mSystems. However, acceptance will not be final until you have adequately addressed the reviewer comments, and revised the manuscript according to the two reviewers' critical and constructive questions.

Preparing Revision Guidelines

Please return the manuscript within 60 days; if you cannot complete the modification within this time period, please contact me. If you do not wish to modify the manuscript and prefer to submit it to another journal, please notify me of your decision immediately so that the manuscript may be formally withdrawn from consideration by mSystems.

Sincerely,

Li Cui

Editor, mSystems

Reviewer comments:

Reviewer #2 (Comments for the Author):

The manuscript by Zheng et al. proposed that ICEs are overlooked "hot" vectors to facilitate ARG propagation from aspects of mobility and pathogenicity through genomic analysis. However, I think the novelty is limited. ICEs have been believed to be the dynamic vectors of the ARG and virulence genes. I have made several suggestions below that in my view might enhance the quality of this manuscript.

#Major:

1. The prediction of ICE boundaries is very important for the analysis of the cargo genes, especially for the insertion site-distal boundaries. This study only simply used the flanking 'DR'-based method; however, lots of the ICEs might carry multiple DR pairs while some ICEs might not have the flanking DR. The ICE boundaries would be validated by the data taken from the chromosomal excision experiments.

2. The method to detect ICEs in the Firmicute genome sequences would be checked as mentioned by the newly published paper (PMID: 36285285).

3. Why did conjugative plasmid and class 1 integron be chosen as the representative MGEs for comparative analysis? The integron is an ARG repository and has no autonomous transfer capability. Notably, the IS element- and transposon-related ARGs are involved in the mosaic structure of ICE or plasmid. Only ICE and plasmid were considered as the ARG carriers may mislead the conclusion.

4. This study only focuses on the ARG profiles on ICEs in different host bacteria. But the different ICEs belonging to the same family might have different ARG profiles. The ARG profile on ICEs would be examined in the diverse strains of the same species, especially for the Enterobacteriaceae strains that carry lots of MGEs.

#Minor:

1. The Conjugative plasmid was defined by Plascad based on the relaxase gene and the T4SS gene clusters. Why the critical factors were not considered? Say, the oriT region and T4CP gene.

2. The color used in Figure 2 is confusing. The same taxon could use the same color.

Reviewer #3 (Comments for the Author):

Zheng et al investigated the MGEs (ICEs, plasmid and integrons) and their associated ARGs and VFs based on the collected bacterial genome data. The data were well analyzed, and the manuscript was well written. However, several issues need to be further administrated. Here I put my comments as follows.

1. English need to be further polished. Some sentences or words should be further well edited. For example, Line 40 "and further to optimize"; Line 49 "in that"; line 64 "to realize"; line "distinguished from" etc.

2. Figure S2 Coverage distribution of the three MGEs encoding different ARG numbers. What do you mean "MGE coverage"?

3. Line 141 what are the shared 34 ARG subtypes?

4. line 143 Different MGEs exhibited the ARG preference. Aminoglycoside resistant genes were prevalent among these three MGEs. The ARGs encoding resistance to trimethoprim and beta-lactam tend to be located on class 1 integrons, while beta-lactam resistance genes were frequently detected on plasmids. I am curious about what factors contribute to the ARG preference on the MGEs? The arg preference on MGEs should be further discussed.

5. Line 121 In this study, the authors focused on the ARGs and VFs on these MGEs. What are the other functional genes carried by these MGEs? Why do these MGEs tend to carry these functional genes? their contribution to bacterial evolution?

6. line 136 the authors found that plasmids and ICEs can carry more than ten ARGs. These ARGs can confer resistance to different antibiotics? What are the characteristics of ARG location on MGEs? For example, aminoglycoside resistance genes (aac(6)-I and aadA) tend to locate on the first gene cassettes..

Zheng et al investigated the MGEs (ICEs, plasmid and integrons) and their associated ARGs and VFs based on the collected bacterial genome data. The data were well analyzed, and the manuscript was well written. However, several issues need to be further administrated. Here I put my comments as follows.

1. English need to be further polished. Some sentences or words should be further well edited. For example, Line 40 “and further to optimize”; Line 49 “in that”; line 64 “to realize”; line “distinguished from” etc.
2. Figure S2 Coverage distribution of the three MGEs encoding different ARG numbers. What do you mean “MGE coverage”?
3. Line 141 what are the shared 34 ARG subtypes?
4. line 143 Different MGEs exhibited the ARG preference. Aminoglycoside resistant genes were prevalent among these three MGEs. The ARGs encoding resistance to trimethoprim and beta-lactam tend to located on class 1 integrons, while beta-lactam resistance genes were frequently detected on plasmids. I am curious about what factors contribute to the ARG preference on the MGEs? The arg preference on MGEs should be further discussed.
5. Line 121 In this study, the authors focused on the ARGs and VFs on these MGEs. What are the other functional genes carried by theses MGEs? Why do these MGEs tend to carry these functional genes? their contribution to bacterial evolution?
6. line 136 the authors found that plasmids and ICEs can carry more than ten ARGs. These ARGs can confer resistance to different antibiotics? What are the characteristics of ARG location on MGEs? For example, aminoglycoside resistance genes (*aac(6')*-I and *aadA*) tend to locate on the first gene cassettes..

Point-by-point Response to Reviewers (mSystems00178-23)

Manuscript ID: mSystems00178-23

Title: Is ICE Hot: A genomic comparative study reveals integrative and conjugative elements (ICEs) as “hot” vectors for the dissemination of antibiotic resistance genes

Authors: Qi Zheng, Liguan Li, Xiaole Yin, You Che, Tong Zhang

Correspondence: Prof. Tong Zhang (zhangt@hku.hk)

Reviewer #2 (Comments for the Author):

The manuscript by Zheng et al. proposed that ICEs are overlooked "hot" vectors to facilitate ARG propagation from aspects of mobility and pathogenicity through genomic analysis. However, I think the novelty is limited. ICEs have been believed to be the dynamic vectors of the ARGs and virulence genes. I have made several suggestions below that in my view might enhance the quality of this manuscript.

#Major:

1. The prediction of ICE boundaries is very important for the analysis of the cargo genes, especially for the insertion site-distal boundaries. This study only simply used the flanking 'DR'-based method; however, lots of the ICEs might carry multiple DR pairs while some ICEs might not have the flanking DR. The ICE boundaries would be validated by the data taken from the chromosomal excision experiments.

Response:

Thanks for raising the concern on approach of ICE detection. In fact, the developer of ICEfinder (1) also noted in the paper that “for various reasons, ICEfinder may not provide the precise boundaries of ICEs”; and regarding the up-to-date genomic ICE-prediction tools, “the accurate delimitation of ICEs still has a long way to go”. Since the performance of ICEfinder was verified against experimentally validated reference ICEs, the identified ICEs using ICEfinder in our study should be reliable.

2. The method to detect ICEs in the Firmicute genome sequences would be checked as mentioned by the newly published paper (PMID: 36285285).

Response:

Thanks for your valuable comment. The total 1022 ICEs extracted from the phylum *Firmicutes* in our study were all validated via the tool ICEscreen (2), which has been added in the “Materials and Methods” section of our manuscript: In addition, ICEs extracted from the phylum *Firmicutes* were verified through ICEscreen (62) with reference to the composite structures of signature proteins dedicated to ICEs located in *Firmicutes* (Line 350-352, Page 18, File “Marked-Up Manuscript”).

3. Why did conjugative plasmid and class 1 integron are chosen as the representative MGEs for comparative analysis? The integron is an ARG repository and has no autonomous transfer capability. Notably, the IS element- and transposon-related ARGs are involved in the mosaic structure of ICE or plasmid. Only ICE and plasmid were considered as the ARG carriers may mislead the conclusion.

Response:

We fully agree with you that transposable elements such as IS or transposons are also significant genetic vectors to transfer ARGs, and it is the concerted activities of these different MGEs that play a pivotal role in the dissemination of ARGs. However, the final decision to exclude transposable elements in our study lies in the following considerations:

(1) Their transferring modes are distinct: conjugative plasmids share the same conjugation machinery as ICEs to promote intercellular mobility, making it meaningful to compare their ARG profiles; whereas transposable elements promote intracellular mobility. Although class 1 integrons are non-autonomous elements, they were compared with ICEs and conjugative plasmids as versatile repositories to accumulate ARGs.

(2) The intact structures of these three MGEs (ICEs, plasmids, and integrons) were extracted using well-developed tools in our study to identify their accessory genes. However, there have been no such well-developed tools so far to extract the intact

structures of prokaryotic transposable elements; instead, the commonly used method to identify transposable elements at present is sequence alignment with well-curated database (3, 4).

(3) Transposable elements may locate in ICEs or conjugative plasmids to conduct intercellular transmission, but these transposable elements as well as class 1 integrons could be regarded as parts of ICEs and plasmids, that is, these ICEs and plasmids were regarded as a whole, and there was no need to discard the transposable element or integron parts on them. Furthermore, as mentioned in our manuscript (Line 378-385, Page 19-20, File “Marked-Up Manuscript”), there existed no specific enrichment of integron-borne ARGs among ICEs and plasmids. In fact, we also identified potential transposable elements from the extracted ICEs and plasmids in our study by sequence alignment against a well-curated MGE database associated with ARGs (BLAST alignment; e-value $\leq 1e-5$, hit similarity $\geq 90\%$, and alignment length coverage $\geq 90\%$) (5). The results indicated that 8% of ICEs and 42% of plasmids carried IS- and transposon-related transposases. Additionally, the versatile interactions of ICEs and plasmids with other MGEs like IS, transposons, or integrons were further discussed to elucidate the high ARG diversity appearing on these two MGEs (Line 289-291, Page 15, File “Marked-Up Manuscript”).

4. This study only focuses on the ARG profiles on ICEs in different host bacteria. But the different ICEs belonging to the same family might have different ARG profiles. The ARG profile on ICEs would be examined in the diverse strains of the same species, especially for the *Enterobacteriaceae* strains that carry lots of MGEs.

Response:

We totally agree with you that MGEs under the same taxon might exhibit different ARG profiles. However, we did not observe obvious occurrence patterns of ARGs on these MGEs at strain level within the same species, probably due to the limited amount of their host genomes. Thus, we focused on their dominant host species to summarize the typical ARG occurrence patterns as well as their co-occurrence patterns with VFs.

#Minor:

1. The conjugative plasmid was defined by Plascad based on the relaxase gene and the T4SS gene clusters. Why the critical factors were not considered? Say, the *oriT* region and T4CP gene.

Response:

Thanks for raising this issue. The confusion might be caused by the ambiguous expression of “T4SS clusters” in our manuscript which actually included T4CP and T4SS. To avoid this probable misunderstanding among future readers, we have replaced these “T4SS clusters” with “T4CP and T4SS” throughout our manuscript. In addition, the core function of Plascad is to classify the identified plasmid sequences into three categories (conjugative, mobilizable, and non-mobilizable) based on the combination of their signature proteins including relaxase, T4CP, and T4SS, i.e., Plascad pays no more attention to *oriT*. Indeed, the performance of Plascad for classification was validated by comparison with an experimentally verified plasmid benchmark dataset (6).

2. The color used in Figure 2 is confusing. The same taxon could use the same color.

Response:

Thanks for your kind reminder. Actually, we had tried to normalize the taxonomic color, but due to certain technical issue, we finally chose to mark the detailed scientific names clearly beside the nodes.

Reviewer #3 (Comments for the Author):

Zheng et al investigated the MGEs (ICEs, plasmids and integrons) and their associated ARGs and VFs based on the collected bacterial genome data. The data were well analyzed, and the manuscript was well written. However, several issues need to be further administrated. Here I put my comments as follows.

1. English needs to be further polished. Some sentences or words should be further

well edited. For example, Line 40 "and further to optimize"; Line 49 "in that"; Line 64 "to realize"; Line 74 "distinguished from" etc.

Response:

Thanks for your careful reading and pointing out these inappropriate expressions. Based on your comment, we have thoroughly revised the whole manuscript, and corrected the inappropriate expressions.

2. Figure S2 Coverage distribution of the three MGEs encoding different ARG numbers. What do you mean "MGE coverage"?

Response:

Thanks for pointing out this issue. "MGE coverage" refers to the proportion of MGEs that encode certain ARG number occupying the total ARGs-carrying MGEs. For example, among the total 560 ARGs-carrying T4SS-type ICEs, 315 ICEs encoded one ARG, then the MGE coverage of ICEs encoding one ARG was $315/560=56.25\%$. Besides, in order to express more explicitly, we have altered the figure caption as "Coverage distribution of the three MGEs encoding different ARG numbers among their total ARGs-carrying ones correspondingly".

3. Line 141 What are the shared 34 ARG subtypes?

Response:

The shared 34 ARG subtypes among the three MGEs (ICEs, plasmids, and integrons) are listed as follows:

1	Aminoglycoside__aac(3)-II
2	Aminoglycoside__aac(6')-I
3	Aminoglycoside__aac(6')-II
4	Aminoglycoside__aadA
5	Aminoglycoside__ant(2'')-I
6	Aminoglycoside__aph(3')-I
7	Aminoglycoside__aph(3'')-I

8	Aminoglycoside__aph(6)-I
9	Beta-lactam__CTX-M-15
10	Beta-lactam__IMP-1
11	Beta-lactam__IMP-14
12	Beta-lactam__Metallo
13	Beta-lactam__NDM-1
14	Beta-lactam__OXA-1
15	Beta-lactam__OXA-2
16	Beta-lactam__PSE-1
17	Beta-lactam__SHV
18	Beta-lactam__TEM-1
19	Chloramphenicol__cat
20	Chloramphenicol__catB
21	Chloramphenicol__cmlA
22	Chloramphenicol__Exporter
23	Macrolide-Lincosamide-Streptogramin__ereA
24	Macrolide-Lincosamide-Streptogramin__mphA
25	Quinolone__qnrB
26	Quinolone__qnrS
27	Rifamycin__arr
28	Tetracycline__tetA
29	Tetracycline__tetB
30	Tetracycline__tetC
31	Tetracycline__tetD
32	Trimethoprim__dfrA1
33	Trimethoprim__dfrA12
34	Trimethoprim__dfrA17

4. Line 143 Different MGEs exhibited the ARG preference. Aminoglycoside resistant

genes were prevalent among these three MGEs. The ARGs encoding resistance to trimethoprim and beta-lactam tend to locate on class 1 integrons, while beta-lactam resistance genes were frequently detected on plasmids. I am curious about what factors contribute to the ARG preference on the MGEs? The ARG preference on MGEs should be further discussed.

Response:

Thanks for pointing out this critical issue. The observation that certain ARGs presented typical preference to specific MGE groups from our study is in line with previous studies (6-9). We fully agree with you that it will be more profound if we could illuminate the underlying causes, and we speculate that this phenomenon may result from external environment of certain intensive antibiotic selective pressure to specific MGE groups making them more likely exposed to corresponding ARGs; or internal molecular mechanisms from these MGEs to preferentially acquire certain ARGs. However, since these speculations lack experimental support, we just briefly prospect it in the “Discussion” section: future work is needed to experimentally evaluate and compare HGT potential of the three MGEs, in particular to illuminate the underlying molecular mechanisms behind characteristic ARG preference of each MGE group (Line 329-332, Page 17, File “Marked-Up Manuscript”).

5. Line 121 In this study, the authors focused on the ARGs and VFs on these MGEs. What are the other functional genes carried by these MGEs? Why do these MGEs tend to carry these functional genes? Their contribution to bacterial evolution?

Response:

Thanks for your constructive comment. In fact, ARGs also tend to co-occur with metal resistance genes (MRGs) on various MGEs as a consequence of widespread metal pressure in the environment to facilitate the co-selection of ARGs and MRGs. Moreover, the co-selection of ARGs and MRGs plays a pivotal role in the development of bacterial antibiotic resistance as well as metal resistance, thus conferring the host bacteria with selective advantages under not only antibiotic pressure but also metal pressure (10). Our study focused on the horizontal gene

transfer among pathogenic bacteria, therefore more attention was paid to the co-occurrence of ARGs and VFs as they possess closer association with pathogens. Meanwhile, MRG profiles of the three identified MGEs (ICEs, plasmids, and integrons) in our study were also analyzed by BLASTP search against a well-curated MRG database ComMet (e-value $\leq 1e-5$, hit similarity $\geq 80\%$, and alignment length coverage $\geq 90\%$) (10). The detailed results are listed as follows:

3761 T4SS-type ICEs			
	MRG Type	ICE Number	ICE Proportion (%/3761)
1	Fe	154	4.1%
2	Zn	136	3.6%
3	Cu	120	3.2%
4	Hg	78	2.1%
5	Sb	66	1.8%
6	Ag	49	1.3%
7	Te	20	0.5%
8	Ni	19	0.5%
9	Au	16	0.4%
10	Se	14	0.4%
11	As	12	0.3%
12	Cr	10	0.3%
13	Mg	5	0.1%
14	Cd	3	0.1%
15	Mn	2	0.1%
16	Pb	2	0.1%
3180 Conjugative Plasmids			
	MRG Type	Plasmid Number	Plasmid Proportion (%/3180)
1	Hg	300	9.4%
2	Sb	235	7.4%

3	Ag	221	6.9%
4	Cu	217	6.8%
5	Ni	148	4.7%
6	Mn	79	2.5%
7	Te	31	1.0%
8	As	25	0.8%
9	Fe	12	0.4%
10	Zn	8	0.3%
11	Pb	3	0.1%
12	Cr	1	0.0%
1008 Class 1 Integrons			
	MRG Type	Integron Number	Integron Proportion (%/1008)
1	Hg	3	0.3%

6. Line 136 The authors found that plasmids and ICEs can carry more than ten ARGs. These ARGs can confer resistance to different antibiotics? What are the characteristics of ARG location on MGEs? For example, aminoglycoside resistance genes (*aac(6')-I* and *aadA*) tend to locate on the first gene cassettes.

Response:

Thanks for raising these professional questions. The ARG structural arrangements of 4 T4SS-type ICEs and 91 conjugative plasmids that carried more than ten ARGs were further analyzed according to the detailed ARG locations on these two MGEs. The results indicated diverse ARG structural arrangements without significant patterns (listed in Excel File “Supplementary Table for Response to Reviewers”).

References

1. Liu M, Li X, Xie Y, Bi D, Sun J, Li J, Tai C, Deng Z, Ou H-Y. 2019. ICEberg 2.0: an updated database of bacterial integrative and conjugative elements. *Nucleic Acids Research* 47:D660-D665.
2. Lao J, Lacroix T, Guédon G, Coluzzi C, Payot S, Leblond-Bourget N, Chiapello H. 2022. ICEscreen: a tool to detect Firmicute ICEs and IMEs, isolated or enclosed in

- composite structures. *NAR Genomics and Bioinformatics* 4:lqac079.
3. Storer JM, Hubley R, Rosen J, Smit AF. 2022. Methodologies for the De novo Discovery of Transposable Element Families. *Genes* 13:709.
 4. Rodriguez M, Makalowski W. 2022. Software evaluation for de novo detection of transposons. *Mobile DNA* 13:14.
 5. Ellabaan MM, Munck C, Porse A, Imamovic L, Sommer MO. 2021. Forecasting the dissemination of antibiotic resistance genes across bacterial genomes. *Nature Communications* 12:2435.
 6. Che Y, Yang Y, Xu X, Břinda K, Polz MF, Hanage WP, Zhang T. 2021. Conjugative plasmids interact with insertion sequences to shape the horizontal transfer of antimicrobial resistance genes. *PNAS* 118:e2008731118.
 7. Johnson CM, Grossman AD. 2015. Integrative and Conjugative Elements (ICEs): What They Do and How They Work. *Annual Review of Genetics* 49:577-601.
 8. Zhang AN, Li L-G, Ma L, Gillings MR, Tiedje JM, Zhang T. 2018. Conserved phylogenetic distribution and limited antibiotic resistance of class 1 integrons revealed by assessing the bacterial genome and plasmid collection. *Microbiome* 6:130.
 9. Che Y, Xia Y, Liu L, Li A-D, Yang Y, Zhang T. 2019. Mobile antibiotic resistome in wastewater treatment plants revealed by Nanopore metagenomic sequencing. *Microbiome* 7:44.
 10. Li L-G, Xia Y, Zhang T. 2017. Co-occurrence of antibiotic and metal resistance genes revealed in complete genome collection. *The ISME Journal* 11:651-662.

August 12, 2023

Prof. Tong Zhang
The University of Hong Kong
Hong Kong
China

Re: mSystems00178-23R1 (Is ICE Hot: A genomic comparative study reveals integrative and conjugative elements (ICEs) as "hot" vectors for the dissemination of antibiotic resistance genes)

Dear Prof. Tong Zhang:

Thank you for submitting your manuscript to mSystems. We have completed our review and I am pleased to inform you that, in principle, we expect to accept it for publication in mSystems. However, acceptance will not be final until you have adequately addressed the reviewer comments.

Preparing Revision Guidelines

Please return the manuscript within 60 days; if you cannot complete the modification within this time period, please contact me. If you do not wish to modify the manuscript and prefer to submit it to another journal, please notify me of your decision immediately so that the manuscript may be formally withdrawn from consideration by mSystems.

Sincerely,

Li Cui

Editor, mSystems

Journals Department
Reviewer comments:

Reviewer #2 (Comments for the Author):

Most of my comments had been considered and the quality of this manuscript was enhanced. The following points would be considered.

1. Conjugative plasmids serve as excellent comparative subjects for analyzing ICEs, but Integrons may be not suitable due to differences in element size, autonomous mobility, and inherent structure. Moreover, Integrons are also distributed on both plasmids and ICEs, especially plasmids, when analyzing the difference and abundance of ARGs among different MGEs, these overlapping elements should be taken into consideration. Maybe the Prophages or other Genomeic islands are more suitable than Integrons. Additionally, I don't agree that whether there are well-developed research tools is the determining factor for selecting the MGEs under analysis.

2. The newly available tool would be tested for ICEs and their boundaries (PMID: 37526274 DOI: 10.1093/nar/gkad644).

Reviewer #3 (Comments for the Author):

The manuscript has been well revised. However, there are several problems regarding the English language as well as data interpretation that I have highlighted below:

Line 127 among the T4SS-type ICEs, there could be many ICE subtypes. Which subtypes of ICEs were abundant in this study? And some novel ICEs can be identified?

Line 168-176 the taxonomic names from bacterial phylum to family should not be in italic.

Line 217 "which was much higher than other species" please revise the description. You mean "which" is the proportion? The proportion is higher than other species? it is hard to understand.

Line 219-222 I can't see the relationship between the mutants of *P. aeruginosa* with your data. In this study, the genes encoding the active efflux system was frequently detected in these genomes?

Line 223-224 line 227 please redescribe the sentences. "tetM appeared predominant on ICEs hosted by"? "most of the two MGEs' host species were shared between each other"

Line 244 You mean the number of VF subtypes?

Line 307-308 the authors mentioned that plasmids are of increased HGT capacity compared to ICEs. What is the range of the rate for horizontal gene transfer of ICEs?

Point-by-point Response to Reviewers (mSystems00178-23R1)

Manuscript ID: mSystems00178-23R1

Title: Is ICE Hot: A genomic comparative study reveals integrative and conjugative elements (ICEs) as “hot” vectors for the dissemination of antibiotic resistance genes

Authors: Qi Zheng, Liguan Li, Xiaole Yin, You Che, Tong Zhang

Correspondence: Prof. Tong Zhang (zhangt@hku.hk)

Reviewer #2 (Comments for the Author):

Most of my comments had been considered and the quality of this manuscript was enhanced. The following points would be considered.

1. Conjugative plasmids serve as excellent comparative subjects for analyzing ICEs, but Integrons may be not suitable due to differences in element size, autonomous mobility, and inherent structure. Moreover, Integrons are also distributed on both plasmids and ICEs, especially plasmids, when analyzing the difference and abundance of ARGs among different MGEs, these overlapping elements should be taken into consideration. Maybe the Prophages or other Genomic islands are more suitable than Integrons. Additionally, I don't agree that whether there are well-developed research tools is the determining factor for selecting the MGEs under analysis.

Response:

We agree with you that research tool is not the determining factor, thus we chose class 1 integrons as the third comparative subjects apart from conjugative plasmids and ICEs mainly in consideration of their different roles to disseminate ARGs: class 1 integrons act as essential ARG reservoirs and rely on conjugative plasmids or ICEs as carriages to conduct intercellular transmission, so we wonder whether these ARG reservoirs present different ARG profiles from their carriages or affect the ARG profiles of their carriages? Interestingly, we found that the three MGEs exhibited distinct ARG profiles, and this finding is of great significance to illuminate MGE-specific potential to facilitate ARG propagation across phylogenetic barriers.

Regarding the several aspects you pointed out for integron analysis, we all have considered and elucidated in our study:

(1) element size: the ARG number per MGE was normalized with their element size and listed in Supplementary Table S2;

(2) inherent structure: ARGs of *qacEΔ1* and *sul1* as inherent integron structures were excluded from integron analysis (Line 379-381, Page 19-20, File “[R2] Marked-Up Manuscript”);

(3) overlapping ARGs: there existed no specific enrichment of integron-borne ARGs among plasmids or ICEs (Line 381-388, Page 20, File “[R2] Marked-Up Manuscript”), meanwhile the detailed integron-borne ARG profiles located on plasmids and ICEs were also listed in Supplementary Table S10.

As for the phages you mentioned, it is controversial whether they disseminate ARGs or not, and a recent study reasserted that ARGs are rarely encoded in phages by means of bioinformatic detection as well as experimental verification with conservative and exploratory thresholds (1). Besides, our lab constructed a database of ~50,000 phages from six WWTPs in Hong Kong (2), but we detected none ARGs from this phage database under BLASTP search criteria of e-value $\leq 1e-5$, hit similarity $\geq 90\%$, and alignment length coverage $\geq 80\%$. Therefore, phages were not considered in our study.

2. The newly available tool would be tested for ICEs and their boundaries (PMID: 37526274 DOI: 10.1093/nar/gkad644).

Response:

Thanks for your valuable advice. We have added this double confirmation in the “Materials and Methods” section of our manuscript: These detected ICEs were double confirmed by a systematic genomic island classification tool AtollGen-CLI (62) according to the combination of their mobility signature proteins (Line 352-354, Page 18, File “[R2] Marked-Up Manuscript”). In detail, the total 3761 T4SS-type ICEs were classified into 3384 (90.0%) “ICE”, 148 “uGI”, 144 “psiICE”, 46 “IE”, and 39

“IME” while the total 403 AICEs were classified into 381 (94.5%) “AICE”, 15 “uGI”, and 7 “IE”. As interpreted in this paper, these classification variations are reasonable and mainly result from missing signatures or low e-value / coverage (3).

Reviewer #3 (Comments for the Author):

The manuscript has been well revised. However, there are several problems regarding the English language as well as data interpretation that I have highlighted below:

Line 127 among the T4SS-type ICEs, there could be many ICE subtypes. Which subtypes of ICEs were abundant in this study? And some novel ICEs can be identified?

Response:

Thanks for pointing out these valuable prospects. Since this study mainly focused on ARG profile comparison among the three MGE groups – ICEs, conjugative plasmids, and class 1 integrons, we did not further analyze ICEs at subtype level; but we concisely discussed potential association between the prominent existence of *tetM* on ICEs and their pivotal subtype of Tn916-like elements (Line 317-322, Page 16-17, File “[R2] Marked-Up Manuscript”).

Line 168-176 the taxonomic names from bacterial phylum to family should not be in italic.

Response:

Thanks for your scientific reminder. It is professional expression that the taxonomic names from bacterial phylum to family are not in italic; however, it has also been recommended to set scientific names at all taxonomic ranks in italic for facilitating their quick recognition in scientific papers (4). Among the recent papers published in mSystems, we notice that there exist both “only genus and species names in italic” (DOI: <https://doi.org/10.1128/msystems.00197-23>, 13 September 2023) and “scientific names at all taxonomic ranks in italic” (DOI: <https://doi.org/10.1128/msystems.00467-23>, 12 September 2023). Maybe we could

check this issue with mSystems editor during format proofreading?

Line 217 "which was much higher than other species" please revise the description. You mean "which" is the proportion? The proportion is higher than other species? it is hard to understand.

Response:

Thanks for raising this issue. We have altered “which” to “their proportions” for clearer expression (Line 218, Page 11, File “[R2] Marked-Up Manuscript”).

Line 219-222 I can't see the relationship between the mutants of *P. aeruginosa* with your data. In this study, the genes encoding the active efflux system was frequently detected in these genomes?

Response:

Thanks for raising this issue. We have split this sentence into two sentences for more explicit expression: The typical multidrug-resistant mutant of *mexEF-oprN* and *mexT* was prevailing among one fifth of ARGs-encoding ICEs embedded on *P. aeruginosa*. Indeed, this multidrug-resistant combination is frequently detected from *P. aeruginosa* to overproduce active efflux system, i.e., the *mexEF-oprN* efflux pump is overexpressed by its activator *mexT* mutation to confer multidrug resistance (38) (Line 221-226, Page 12, File “[R2] Marked-Up Manuscript”).

Line 223-224 Line 227 please redescribe the sentences. "*tetM* appeared predominant on ICEs hosted by"? "most of the two MGEs' host species were shared between each other"

Response:

Thanks for raising this issue. We have modified these two sentences (Line 226-227 and Line 230-231, Page 12, File “[R2] Marked-Up Manuscript”).

Line 244 You mean the number of VF subtypes?

Response:

The percentage number here is proportion of MGE number. In detail, among the total 3761 T4SS-type ICEs, 745 (20%) ICEs carried VFs with 132 VF subtypes; whereas among the total 3180 conjugative plasmids, 436 (14%) plasmids carried VFs with 52 VF subtypes. This information was also listed in Supplementary Table S8.

Line 307-308 the authors mentioned that plasmids are of increased HGT capacity compared to ICEs. What is the range of the rate for horizontal gene transfer of ICEs?

Response:

This reference paper drew the conclusion that plasmids are of increased HGT capacity than ICEs mainly by comparing their genetic organizations – plasmids are more likely to encode ARGs while ICEs encode more metabolism-related genes (5). Besides, the transfer rate of ICEs was summarized by a review paper as follows (6):

Table 3. Summary of Features Observed in Model ICE Families

ICE family prototypes	Originally described in	Common hotspot for integration	Experimentally determined systems ^a	Transfer rate (per donor)	Number of experimentally tested family members ^c	Refs
SXT/R391	Vibrio cholerae	Into the 5' end of a prfC gene	Eex, Par, Rep, Rmo, T/At	1×10^{-4}	81	[42,95–99]
Tn916	Enterococcus faecalis	Many different chromosomal regions	Rep	10^{-4} to 10^{-7}	58	[100–102]
ICEclc	Pseudomonas knackmussii	Into the 3' end of a tRNA ^{Gly} gene	Par, Rep	1×10^{-2}	9	[43,103–105]
ICESt1/ICES13	Streptococcus thermophilus	Into the 3' end of a fda gene	Rmo	3.4×10^{-6}	2	[106–108]
ICEBs1	Bacillus subtilis	Into the 3' end of a tRNA ^{Leu} gene	Eex, Rep	1×10^{-2b}	1	[48,109,110]

^aEex, entry exclusion; Par, partition; Rep, replication; Rmo, restriction–modification; T/At, toxin–antitoxin.

^bTransfer rate drops by about 50-fold when the ICE is transferred into recipient cells that already contain ICEBs1 [48].

^cData retrieved from ICEberg, accessed on the 15 April 2020'.

References

1. Enault F, Briet A, Bouteille L, Roux S, Sullivan MB, Petit M-A. 2017. Phages rarely encode antibiotic resistance genes: a cautionary tale for virome analyses. *The ISME Journal* 11:237-247.
2. Chen Y, Wang Y, Paez-Espino D, Polz MF, Zhang T. 2021. Prokaryotic viruses impact functional microorganisms in nutrient removal and carbon cycle in wastewater treatment plants. *Nature Communications* 12:5398.
3. Audrey B, Cellier N, White F, Jacques P-É, Burrus V. 2023. A systematic approach to classify and characterize genomic islands driven by conjugative mobility using protein signatures. *Nucleic Acids Research* 51:8402-8412.
4. Thines M, Aoki T, Crous PW, Hyde KD, Lücking R, Malosso E, May TW, Miller AN, Redhead SA, Yurkov AM. 2020. Setting scientific names at all taxonomic ranks in italics facilitates their quick recognition in scientific papers. *IMA Fungus* 11:25.

5. Cury J, Oliveira PH, de la Cruz F, Rocha EP. 2018. Host Range and Genetic Plasticity Explain the Coexistence of Integrative and Extrachromosomal Mobile Genetic Elements. *Molecular Biology and Evolution* 35:2230-2239.
6. Botelho J, Schulenburg H. 2021. The Role of Integrative and Conjugative Elements in Antibiotic Resistance Evolution. *Trends in Microbiology* 29:8-18.

October 14, 2023

Prof. Tong Zhang
The University of Hong Kong
Hong Kong
China

Re: mSystems00178-23R2 (Is ICE Hot: A genomic comparative study reveals integrative and conjugative elements (ICEs) as "hot" vectors for the dissemination of antibiotic resistance genes)

Dear Prof. Tong Zhang:

I am pleased to inform you that your manuscript has been accepted, and I am forwarding it to the ASM Journals Department for publication. For your reference, ASM Journals' address is given below. Before it can be scheduled for publication, your manuscript will be checked by the mSystems production staff to make sure that all elements meet the technical requirements for publication. They will contact you if anything needs to be revised before copyediting and production can begin. Otherwise, you will be notified when your proofs are ready to be viewed.

If you would like to submit a potential Featured Image, please email a file and a short legend to msystems@asmusa.org. Please note that we can only consider images that (i) the authors created or own and (ii) have not been previously published. By submitting, you agree that the image can be used under the same terms as the published article. File requirements: square dimensions (4" x 4"), 300 dpi resolution, RGB colorspace, TIF file format.

We recognize that the video files can become quite large, and so to avoid quality loss ASM suggests sending the video file via <https://www.wetransfer.com/>. When you have a final version of the video and the still ready to share, please send it to mSystems staff at msystems@asmusa.org.

Sincerely,

Li Cui
Editor, mSystems

Journals Department
E-mail: mSystems@asmusa.org